# High-throughput screens using photo-highlighting discover BMP signaling in mitochondrial lipid oxidation

Yong Yu[1,2], Ayse Sena Mutlu[3], Harrison Liu[4] & Meng C. Wang [1,2,3]

High-throughput screens at microscopic resolution can uncover molecular mechanisms of cellular dynamics, but remain technically challenging in live multicellular organisms. Here we present a genetic screening method using photo-highlighting for candidate selection on microscopes. We apply this method to stimulated Raman scattering (SRS) microscopy and systematically identify 57 *Caenorhabditis elegans* mutants with altered lipid distribution. Four of these mutants target the components of the Bone Morphogenetic Protein (BMP) signaling pathway, revealing that BMP signaling inactivation causes exhaustion of lipid reserves in somatic tissues. Using SRS-based isotope tracing assay to quantitatively track lipid synthesis and mobilization, we discover that the BMP signaling mutants have increased rates of lipid mobilization. Furthermore, this increase is associated with the induction of mitochondrial β-oxidation and mitochondrial fusion. Together these studies demonstrate a photo-highlighting microscopic strategy for genome-scale screens, leading to the discovery of new roles for BMP signaling in linking mitochondrial homeostasis and lipid metabolism.

[1] Huffington Center on Aging, Baylor College of Medicine, Houston, TX 77030, USA. [2] Department of Molecular and Human Genetics, Baylor College of Medicine, Houston, TX 77030, USA. [3] Program in Developmental Biology, Baylor College of Medicine, Houston, TX 77030, USA. [4] Graduate Program in Bioengineering, University of California, San Francisco and University of California, Berkeley, San Francisco, CA 94143, USA. Correspondence and requests for materials should be addressed to M.C.W. (email: wmeng@bcm.edu)

The ability to study whole organisms makes it possible to study complex in vivo processes under physiological conditions that cannot be replicated in vitro or in cell culture systems. High-throughput genetic screens using stereoscopes have identified various genes that regulate animal development, physiology, and behavior in multicellular organisms, such as *Drosophila*, *C. elegans*, and zebrafish[1, 2]. On the other hand, these multicellular organisms can also be advantageous to elucidate the molecular basis of cellular/subcellular structures and their functions in the context of physiological cell–cell interactions. In particular, recently advanced genome editing approaches make it easy to tag endogenous proteins for visualizing their spatiotemporal distribution in living multicellular organisms. However, many cellular/subcellular phenotypes are only visible under high-resolution microscopes, and compared to high-copy transgenes, endogenous levels of gene expression are often low and require high-sensitivity detection. Integrating high-throughput screening with sophisticated high-resolution, high-sensitivity microscopy analysis is challenging because it usually requires specialized instrumentation for sample manipulation[3, 4]. In order to make whole-organism high-throughput microscopic screening more accessible, we sought to develop an approach that, when imaging cellular phenotypes, marks positive candidates by easily identifiable fluorescence signals (photo-highlighting). This photo-highlighting strategy can be readily compatible with any types of fluorescent and non-fluorescent microscopy systems for large-scale genetic screens of various cellular/subcellular phenotypes with high resolution and sensitivity.

Stimulated Raman scattering (SRS) microscopy is a non-fluorescent laser-scanning optical system that can selectively image the vibration of a specific chemical group[5, 6]. It has been proven a powerful technique for quantitatively visualizing lipid molecules with high spatial and temporal resolution in living cells and organisms[7], leading to the discovery of new genes regulating lipid metabolism[8]. Lipid metabolism is fundamental for various cellular responses, and its dysfunction results in such human diseases as metabolic disorders, neurodegenerative diseases, and cancers. As in proteins, the physiological and pathological activities of lipid molecules are tightly associated with their spatial distribution and temporal dynamics[9, 10]. However the molecular mechanisms underlying the spatiotemporal regulation of lipid molecules remain poorly understood. Therefore we applied the photo-highlighting strategy to harness the power of high-throughput genetic screens and SRS microscopy, and systematically searched for genetic factors that regulate the spatial distribution of lipid molecules at the whole-organism level. Through analyzing the identified mutants from the screen, we discovered the crucial role of bone morphogenetic protein (BMP) signaling in regulating somatic lipid mobilization and the associated mechanism through tuning mitochondrial dynamics.

## Results

### Photo-highlighting with photoconvertible fluorescent proteins.
To develop an approach for marking selective candidates on microscopes (Fig. 1a), we chose photoconvertible fluorescent proteins as a highlighter, which can be switched from one color to another by light in an irreversible manner. For applying this approach in *C. elegans*, we generated a transgenic strain *Is[rab-3p::mEosFP]* that expresses in all neurons the photo-convertible fluorescent protein mEosFP[11], which can irreversibly change color from green to red with 405 nm illumination. To characterize the performance of photo-highlighting, we

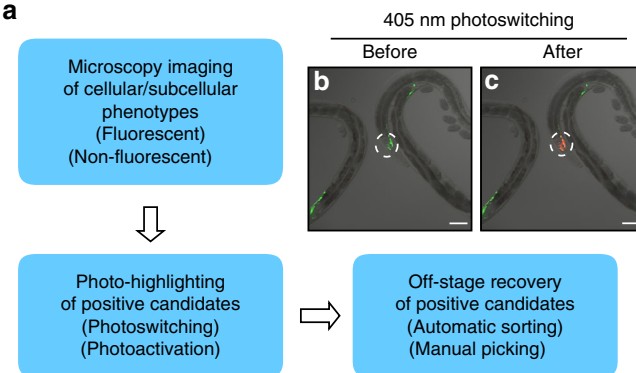

**Fig. 1** Photo-highlighting strategy for microscopic screening. **a** The overview of the photo-highlighting strategy. **b**, **c** The head region of one transgenic worm expressing pan-neuronal mEosFP is photoswitched from *green* **b** to *red* **c** with 405 nm illumination. Photo-highlighted areas circled by dashed lines. *Scale bar* = 50 μm

focused a 405 nm laser to the head region of these transgenic worms under a confocal fluorescence microscope (Fig. 1b). After 15 s of photoactivation, the target worm was brightly marked in red (Fig. 1c). The red fluorescence lasted at least 6 h, and the target worm was easily recognized and rapidly recovered from more than 1000 green worms under a fluorescent stereoscope. Therefore, this photo-highlighting strategy provides an effective way to mark positive candidates when imaging cellular/subcellular phenotypes on microscopes. Following microscopic imaging, these candidates can then be either sorted automatically with a commercial fluorescence-based sorter or manually recovered under a stereoscope. Importantly, given the low hit-rate of genetic mutagenesis screens (typically < 1 out of 1000) and hence the small number of candidates, manual off-microscope recovery will not limit the screening throughput.

### Genome-scale SRS microscopic screens with photo-highlighting.
To demonstrate a real genome-scale screen, we implemented this photo-highlighting strategy on an SRS microscope built on a commercial laser-scanning confocal microscope, and conducted a forward genetic screen to search for genes regulating lipid distribution (Fig. 2a). F2 worms from ethyl methanesulfonate (EMS) mutagenesis were immobilized in 3D printed multi-well chambers (~120 worms/well, Supplementary Fig. 1) and imaged for their lipid phenotypes by SRS microscopy (Fig. 2a and Supplementary Fig. 2a). Once a mutant with altered lipid distribution was visually identified, we switched on the 405 nm laser and focused it at the head region to photo-highlight the mutant (Fig. 2a and Supplementary Fig. 2b). Following the screening of every ~2000 worms, the imaging chamber was moved to a fluorescent stereoscope, where highlighted worms were identified and recovered using a mouth pipette (Fig. 2a and Supplementary Fig. 2c). Using this approach, we screened ~50,000 worms within 40 h and identified 240 mutants (Fig. 2b). Compared to canonical methods (~5 min to image and recover one worm), this new approach is around 100 times faster to complete a screen at the same scale.

Among these candidates, 106 mutants were sterile or dauer constitutive (Fig. 2b). The other 57 fertile mutants were validated through retests and classified into five groups (Class I–V) based on their distinctive alterations in lipid phenotypes (Fig. 2c, d). In *C. elegans*, the accumulation of lipids can be predominantly detected in both somatic tissues, including the intestine and hypodermis, and oocytes in the germline (Fig. 2c). We found that

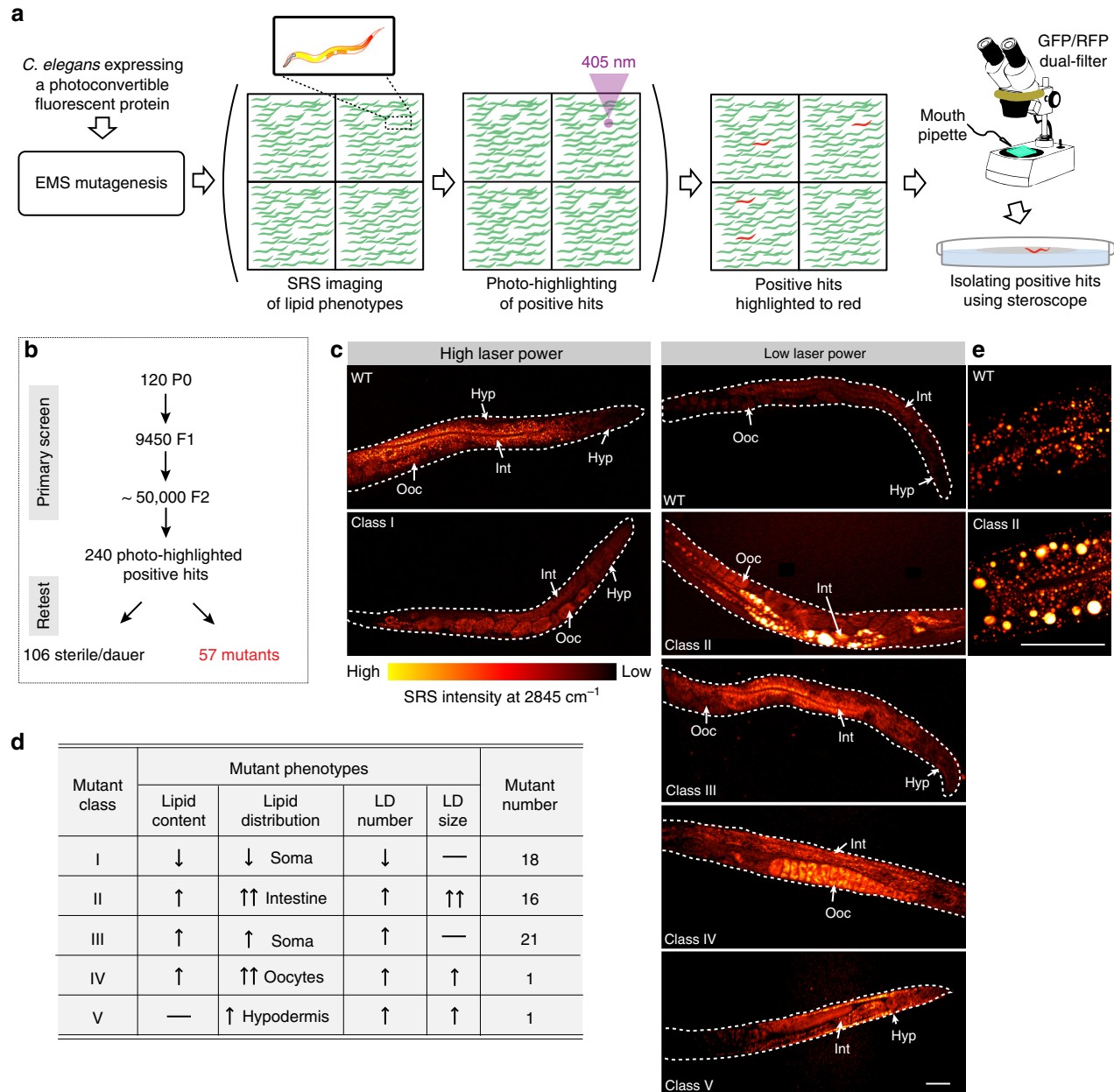

**Fig. 2** High-throughput SRS microscopy screens using photo-highlighting. **a** The methodological scheme of SRS-based forward genetic screens using photo-highlighting. **b** The working flow of primary screens and subsequent retests to identify mutants with altered lipid distribution phenotypes. **c**, **d** 57 mutants were identified with altered lipid distribution and categorized into five classes. Age synchronized, 1-day-old adult worm were imaged. Example SRS images are shown in **c**. *Dashed lines* indicate the boundary of animals. High laser power: 600 mW pump/OPO (816 nm), 400 mW Stokes/IR (1064 nm); low laser power: 200 mW pump/OPO (816 nm), 400 mW Stokes/pump (1064 nm). *Scale bar = 50 μm*; Hyp, hypodermis; Int, intestine; LD, lipid droplets; Ooc, oocytes. **e** Example, high-resolution images show enlarged lipid droplets in the intestine of class II mutants. *Scale bar = 50 μm*

the 18 mutants in Class I have decreased levels of lipid content in somatic tissues, but not in the germline (Fig. 2c, d). In contrast, the 16 and 21 mutants in Class II and Class III, respectively, show increased levels of somatic lipid content (Fig. 2c, d). Interestingly, one mutant in Class IV has abnormal lipid accumulation in oocytes (Fig. 2c, d). Except for this mutant, all the others with abnormal oocyte lipid accumulation are sterile. In the mutant of Class V, the level of lipid content is increased in the hypodermis, but remains unaffected in the intestine (Fig. 2c, d). These results reveal differential regulation of lipid accumulation in distinct tissues.

Moreover, thanks to the subcellular resolution of SRS microscopy, we were able to characterize the mutants that affect the morphology of lipid droplets, specialized cellular organelles for neutral lipid storage. Especially, the 16 mutants in Class II have dramatically enlarged lipid droplets in the intestine (Fig. 2d, e), suggesting their significance in regulating lipid droplet expansion.

**Discovery of BMP signaling in regulating lipid metabolism.** We further characterized four newly isolated mutants in Class I, which show reduced lipid storage and body size, via SNP mapping[12] and next-generation sequencing[13]. Interestingly all four mutants were mapped into the BMP signaling pathway[14, 15], affecting three *C. elegans* Smad proteins, SMA-2, SMA-3, and SMA-4 (Fig. 3a). Mutations in *sma-2(rax5)*, *sma-4(rax3)*,

and *sma-4(rax10)* lead to codon changes (Fig. 3a), while the mutation in *sma-3(rax7)* results in an alternative splicing error and a frame shift (Fig. 3a). All these mutants decrease the number of lipid droplets and the level of lipid content (Fig. 3b, c). In *C. elegans*, the BMP/sma signaling pathway consists of the DBL-1 ligand, the SMA-6 and DAF-4 receptors, the SMA-2 and SMA-3 R-Smads, the SMA-4 Co-Smad, and the SMA-9 transcription factor (Fig. 3d). We found that mutants defective in all these different components of the BMP signaling pathway exhibit similar phenotypes in reducing lipid droplets and lipid content (Fig. 3c), and the reduction in lipid content were further confirmed using fixation-based Oil Red O staining (Supplementary Fig. 3). In contrast, the *daf-1(m40)*, *daf-3(mgDf90)*, *daf-5(e1386)*, *daf-7(e1372)*, *daf-8(e1393)*, and *daf-14(m77)* mutants of the TGF-β/dauer signaling pathway[16] have no such effects (Fig. 3c), suggesting the specificity of BMP/sma signaling in regulating lipid metabolism.

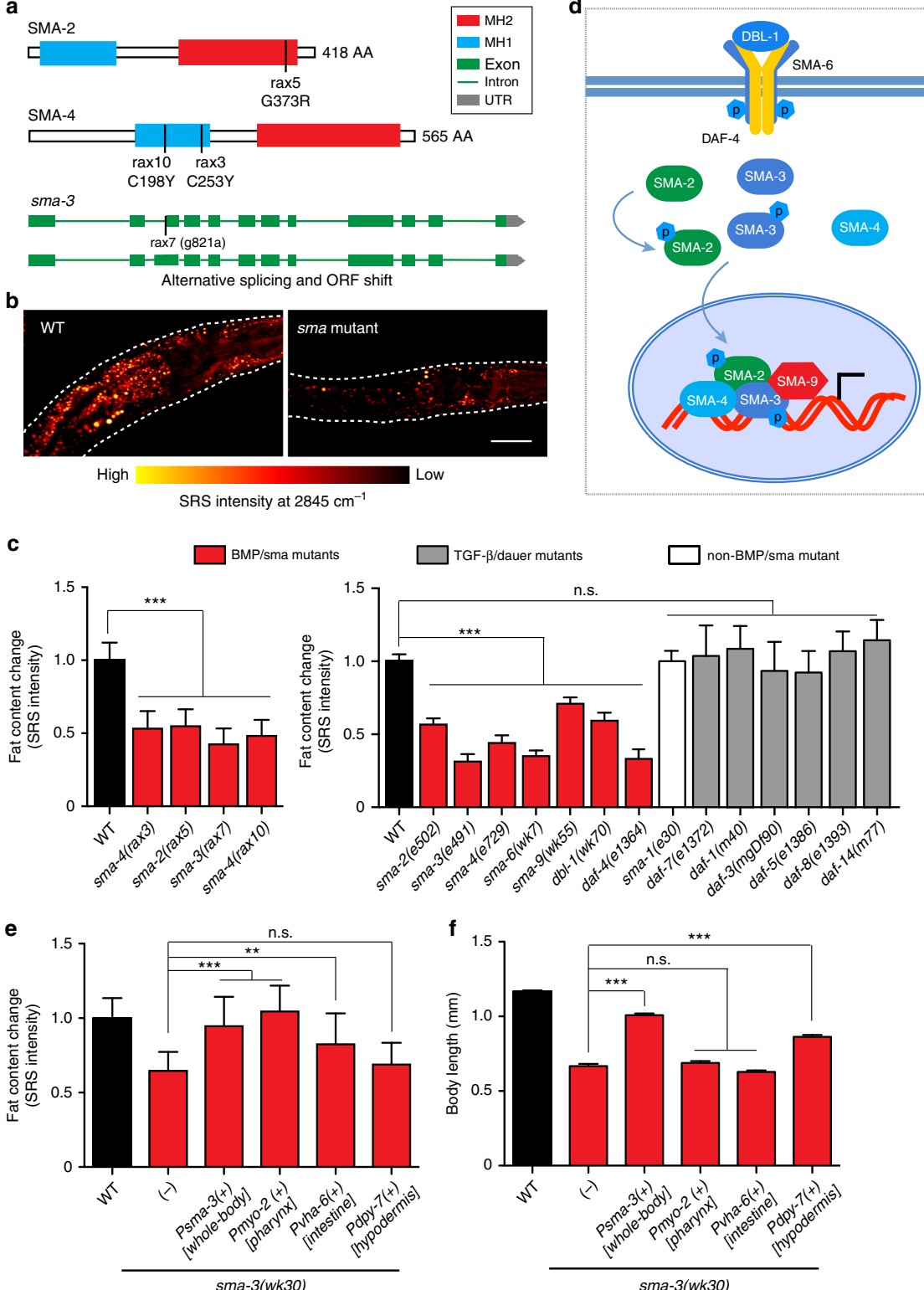

We next examined whether the lipid phenotypes of the BMP signaling mutants are due to alterations in their feeding behaviors and/or physical activities. We measured the food intake rate, defecation cycling duration, and locomotion activity, and found no significant changes in these mutants compared to wild-type worms (Supplementary Fig. 4). Mutations of the BMP signaling pathway are previously known to decrease the body size of worms[16]. To test whether the decreased lipid level is simply a result of small body size, we first examined a non-BMP small mutant, *sma-1(e30)*[17] and detected no alteration in lipid phenotypes (Fig. 3c), suggesting no direct correlation between reduced body size and lipid levels. Furthermore, hypodermal restoration of *sma-3* expression rescued the small body size of the *sma-3* mutant, without correcting the lipid phenotypes (Fig. 3e, f and Supplementary Fig. 5). In contrast, restoration of *sma-3* in pharyngeal muscle or intestine fully or partially rescued lipid reduction, respectively, but neither increased the body size (Fig. 3e, f and Supplementary Fig. 5). Thus, BMP signaling actively regulates organism lipid metabolism, through a mechanism independent of its action on body size.

**BMP links lipid mobilization and mitochondrial homeostasis.** Next, we investigated the mechanisms by which BMP signaling components regulate cellular lipid metabolism. Lipid storage in cells is a highly dynamic process, which is balanced between lipid synthesis/incorporation and mobilization at lipid droplets. To track cellular lipid flux, we applied isotope-labeling coupled SRS microscopic imaging[7]. For measuring the rate of lipid synthesis, we supplemented worms with deuterium-labeled oleic acids and tracked the accumulation of newly synthesized lipids carrying deuterium labeling (C-D) over time (Fig. 4a). Conversely, to determine the rate of lipid mobilization, worms were first supplemented with deuterium-labeled oleic acids over 24 h, and the labeled worms were then transferred to unlabeled plates for tracing the decay of C-D signals over time (Fig. 4b). Using these assays, we discovered that lipid synthesis/incorporation remains unaffected in the BMP signaling pathway mutants (Fig. 4c and Supplementary Fig. 6a); however, lipid mobilization is significantly accelerated in these mutants (Fig. 4d and Supplementary Fig. 6b).

To investigate the mechanism by which BMP signaling regulates lipid mobilization, we compared the expression levels of a series of metabolic genes between wild type and the *sma* mutant worms (Fig. 5a). Interestingly, we discovered that both acyl-CoA synthetase (*acs*) and carnitine palmitoyl transferase (*cpt*) genes involved in mitochondrial fatty acid β-oxidation are induced in the *sma* mutants (Fig. 5a, b). Moreover, the loss-of-function mutation of *acs-2(ok2457)* suppresses the decreased lipid level in the *sma* mutant (Fig. 5c), suggesting that BMP signaling regulates lipid mobilization via controlling

mitochondrial β-oxidation. On the other hand, the expression levels of genes involved in lipolysis, TCA cycle, or de novo fatty acid synthesis are not affected by the *sma* mutant (Fig. 5a).

To gain more insights into the effect of BMP signaling on mitochondria, we examined mitochondrial morphology using mito-GFP reporters in the *sma* mutants. We found that mitochondrial fusion is increased, as evident by more connected tubular structures (Fig. 6a). Interestingly, restoration of *sma-3* expression in the pharyngeal muscle or intestine of the *sma-3* mutant can fully or partially rescue the induced mitochondrial fusion, respectively; however restoration of *sma-3* expression in the hypodermis has no such effects (Fig. 6b). These results suggest the pharyngeal activity of BMP signaling plays a predominant role in regulating intestinal fat storage and mitochondrial dynamics cell non-autonomously, and intestinal BMP signaling also makes some contributions cell-autonomously. The morphology of mitochondrial network is governed by a balance between organelle fusion and fission[18, 19], and *fzo-1* encodes *C. elegans* mitofusin essential for mitochondrial fusion[20]. Using the *fzo-1 (tm1133)* mutant, we confirmed that blocking mitochondrial fusion suppresses the decreased lipid level in the *sma* mutant (Fig. 6c), suggesting that BMP signaling regulates lipid mobilization through tuning mitochondrial dynamics. Interestingly, neither the *acs-2* nor the *fzo-1* mutant rescues the small body size of the *sma* mutant (Figs. 5d and 6d), supporting distinct molecular mechanisms underlying BMP signaling in the regulation of lipid metabolism and body size. Together, these results reveal the specific and crucial role of BMP signaling in orchestrating mitochondrial metabolic homeostasis to control lipid catabolism (Fig. 6e).

## Discussion

In summary, we presented a high-throughput microscopy-based phenotypic screening approach using photo-highlighting for candidate selection. This microscopic screening approach is orders of magnitude faster than canonical methods, and enables systematic characterization of previously unknown cellular/sub-cellular phenotypes using either fluorescent or non-fluorescent microscopy systems. In our studies, we have demonstrated this approach for a large-scale forward genetic screen on a laser-scanning SRS microscope. From the screen, we have isolated a series of mutants affecting lipid distribution between tissues and altering organellar morphology of lipid droplets. This demonstration exemplifies the easy adaption of our method for a wide range of microscopy systems and its application to many optically transparent multicellular organisms, thus making inroad into new gene discovery.

In particular, we discovered BMP signaling as a new regulatory pathway of lipid metabolism in *C. elegans*. Mutations of all the components in this pathway, ranging from the membrane

**Fig. 3** Discovery of BMP signaling in cell non-autonomous regulation of lipid metabolism. **a** Four mutants from the screen are defective in *sma-2*, *sma-3*, and *sma-4*. **b** Example SRS images reveal that the *sma* mutants have reduced numbers of lipid droplets and levels of lipid content, compared to wild type. *Dashed lines* indicate the boundary of worms, scale bar = 50 μm. **c** Quantification using SRS signal intensity show that the levels of lipid content are decreased in the BMP/*sma* mutants, but not in the TGF-β/dauer mutants or in the non-BMP *sma* mutant. Age synchronized, 1-day-old adult worms were imaged. ***$P <$ 0.001, n.s. $P >$ 0.05, $n =$ 35 for WT, $n =$ 17 for *rax3*, $n =$ 21 for *rax5*, $n =$ 24 for *rax7*, $n =$ 16 for *rax10*, $n =$ 28 for *e502*, $n =$ 22 for *e491*, $n =$ 26 for *e729*, $n =$ 44 for *wk7*, $n =$ 24 for *wk55*, $n =$ 21 for *wk70*, $n =$ 25 for *e1364*, $n =$ 16 for *e30*, $n =$ 20 for *e1372*, $n =$ 72 for *m40*, $n =$ 69 for *mgDf90*, $n =$ 89 for *e1386*, $n =$ 68 for *e1393*, and $n =$ 75 for *m77*. **d** Diagram of the BMP/SMA signaling pathway in *C. elegans*. **e** Decreased lipid content by the *sma-3* mutation is rescued fully by expressing *sma-3* using its endogenous promoter or the *myo-2* (pharyngeal muscle) promoter, partially by using the *vha-6* (intestine) promoter, but not by using the *dpy-7* (hypodermis) promoter. Representative SRS images are shown in Supplementary Fig. 5. Age synchronized, 1-day-old adult worms were imaged. ***$P <$ 0.001, **$P <$ 0.01, n.s. $P >$ 0.05, $n =$ 54 for WT, $n =$ 44 for *sma-3* mutant, $n =$ 24 for *Psma-3*( + ), $n =$ 42 for *Pmyo-2*( + ), $n =$ 50 for *Pvha-6*( + ), and $n =$ 46 for *Pdpy-7*( + ). **f** The small body size of the *sma-3* mutant is rescued by expressing *sma-3* using its endogenous promoter or the *dpy-7* (hypodermis) promoter, but not by using the *myo-2* (pharyngeal muscle) or the *vha-6* (intestine) promoter. ***$P <$ 0.001, n.s. $P >$ 0.05, $n =$ 17 for WT, $n =$ 20 for *sma-3* mutant, $n =$ 13 for *Psma-3*( + ), $n =$ 15 for *Pmyo-2*( + ), $n =$ 18 for *Pvha-6*( + ), and $n =$ 21 for *Pdpy-7*( + ). In all bar charts, data represent mean ± s.d., $P$ value is determined by a Student's *t*-test (unpaired, two-tailed)

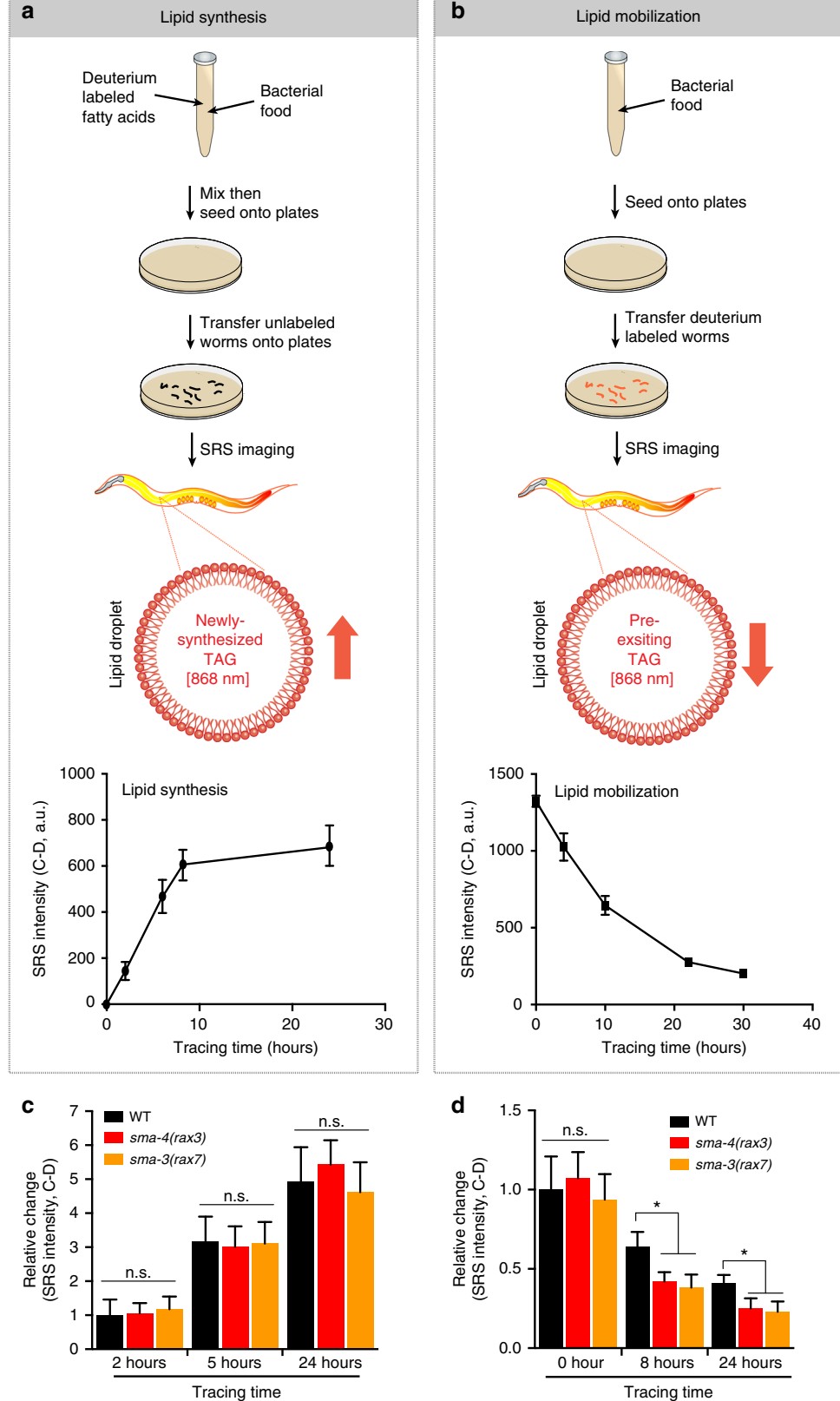

**Fig. 4** Isotope-labeling coupled SRS (iSRS) reveals the regulation of lipid mobilization by BMP signaling. **a**, **b** Diagram of iSRS assays to determine the rate of lipid synthesis **a** and lipid mobilization **b** using deuterium-labeled fatty acids. $n = 16$. **c** The BMP/*sma* mutants do not affect lipid synthesis, n.s. $P > 0.05$, $n = 16$. **d** The BMP/*sma* mutants accelerate lipid mobilization, n.s. $P > 0.05$, *$P < 0.05$, $n = 16$. In all bar charts, data represent mean ± s.d., $P$ value is determined by a Student's *t*-test (unpaired, two-tailed)

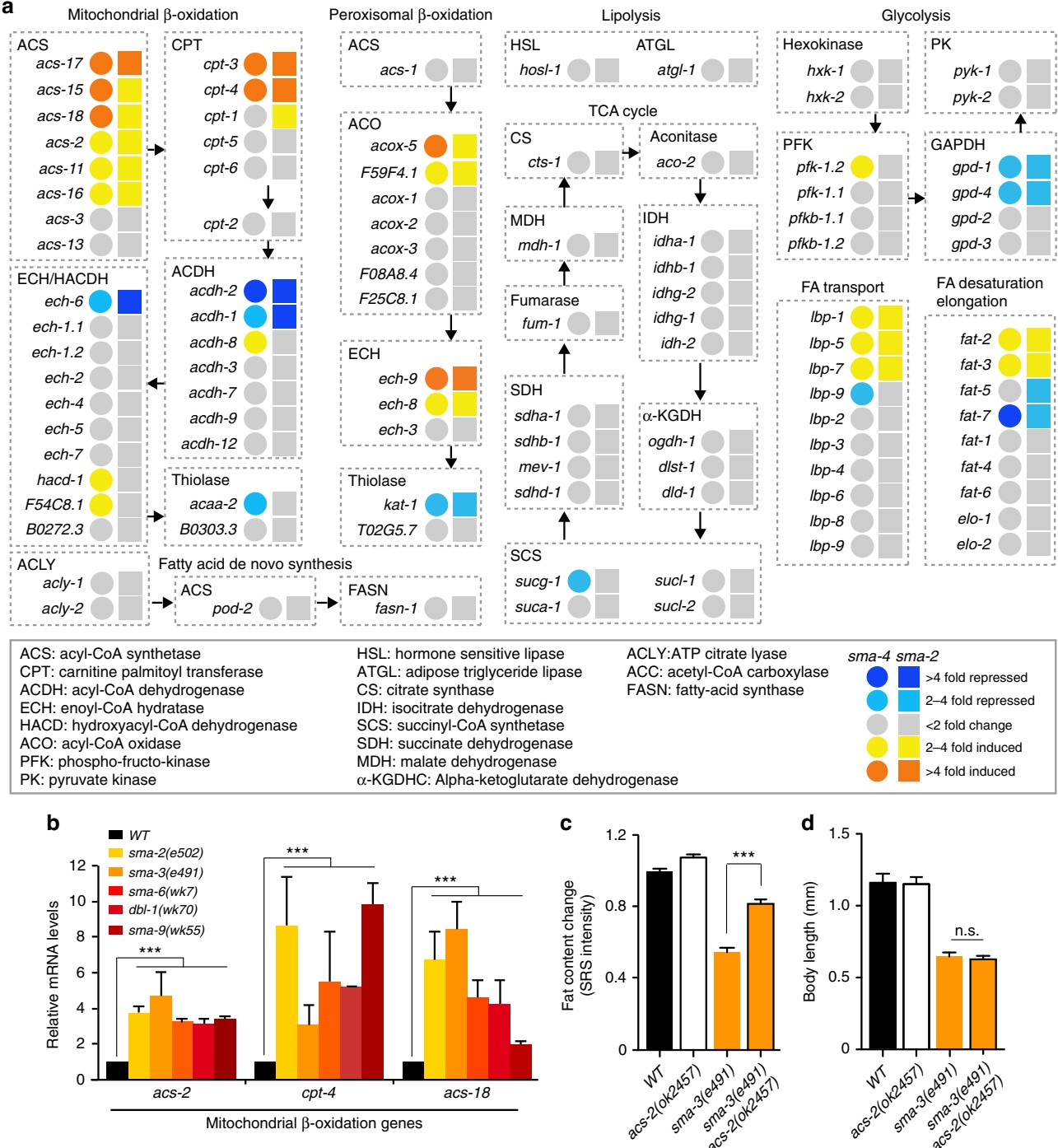

**Fig. 5** BMP signaling regulates mitochondrial β-oxidation to control lipid metabolism. **a** Metabolic gene expression profiles between wild type and the BMP/*sma* mutants. **b** Mitochondrial β-oxidation genes are induced in the *sma* mutants. Error bars represent s.d., ***$P < 0.001$, Student's *t*-test (unpaired, two-tailed), $N = 3$. **c** The loss-of-function mutant of *acs-2(ok2457)* significantly increases the lipid content level of the *sma-3* mutant. *Error bars* represent standard error (s.e.), ***$P < 0.001$, Student's *t*-test (unpaired, two-tailed), $n = 87$ for WT, $n = 74$ for *acs-2*, $n = 57$ for *sma-3*, $n = 63$ for *acs-2;sma-3*. **d** *acs-2* inactivation does not suppress the decrease in body size of the *sma-3* mutant. *Error bars* represent s.e., n.s. $P > 0.05$, Student's *t*-test (unpaired, two-tailed), $n = 39$ for WT, $n = 29$ for *acs-2*, $n = 34$ for *sma-3*, and $n = 29$ for *sma-3;acs-2*

receptor to the transcription factor, lead to decreased lipid accumulation in somatic tissues of adult worms. This regulation acts in metabolic active tissues, predominantly in pharyngeal muscle with a minor effect in the intestine, which is distinct to the previously known function of BMP signaling in the hypodermis to control body size[21, 22]. Pharynx and intestine, together with the rectum, form the digestive tract of *C. elegans*. Even though

pharynx has been thought as an organ solely for the grinding and transportation of bacteria to the intestine, the anatomy of the pharynx suggests that it could be an active signaling hub as well[23]. First, in the posterior, six valve cells connect the last pharyngeal muscular cell to the first pair of intestinal cells. Thus, the pharynx can be considered as a continuum of the anterior intestine. Second, the basal surface of the pharynx is lined by basal lamina,

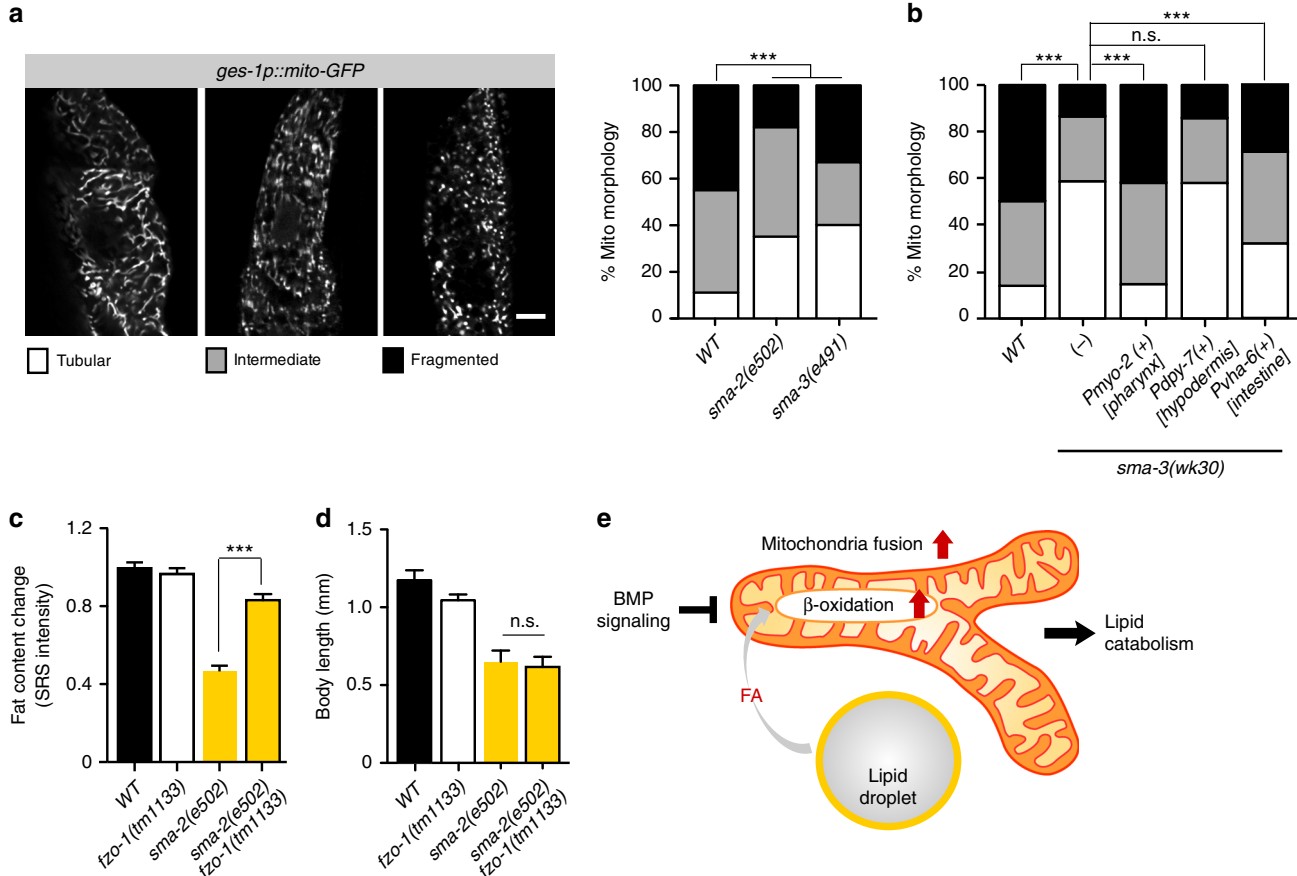

**Fig. 6** BMP signaling regulates mitochondrial dynamics cell non-autonomously. **a** Mitochondrial morphology was analyzed using mito-GFP in intestinal cells (*zcIs17[ges-1p::mitoGFP]*), and classified as tubular, intermediate, and fragmented (representative images are shown, *scale bar* = 10 μm). The BMP/*sma* mutants display significantly increased levels of tubular mitochondrial morphology, compared to wild type worms, ***$P < 0.001$, $\chi^2$ test, $n = 100$. **b** Increased levels of tubular mitochondrial morphology in the *sma-3* mutant is rescued fully by expressing *sma-3* using the *myo-2* (pharyngeal muscle) promoter, partially by the *vha-6* (intestine) promoter, but not by using the *dpy-7* (hypodermis) promoter. ***$P < 0.001$, n.s. $P > 0.05$, $\chi^2$ test, $n = 100$. **c** The loss-of-function mutant of *fzo-1(tm1133)* significantly increases the lipid content level of the *sma-2* mutant. *Error bars* represent s.e., ***$P < 0.001$, Student's *t*-test (unpaired, two-tailed), $n = 41$ for WT, $n = 68$ for *fzo-1*, $n = 35$ for *sma-2*, $n = 74$ for *fzo-1;sma-2*. **d** The *fzo-1* mutant does not rescue the small body size of the *sma-2* mutant. *Error bars* represent s.e., n.s. $P > 0.05$, Student's *t*-test (unpaired, two-tailed), $n = 22$ for WT, $n = 28$ for *fzo-1*, $n = 14$ for *sma-2*, and $n = 16$ for *sma-2;fzo-1*. **e** The model of lipid catabolism regulation by BMP signaling. BMP signaling inhibits mitochondrial β-oxidation and mitochondria fusion. When BMP signaling is blocked, the induction of mitochondrial β-oxidation and fusion leads to increased lipid catabolism

which is in contact with the pseudocoelomic cavity. Therefore, BMP signaling in the pharynx might either directly communicate with the intestine via the connecting valve cells or there may be indirect endocrine signaling between the pharynx and rest of the body including the intestine via the pseudocoelom.

The essential roles of BMP signaling in embryonic patterning and organ formation are well known in *Drosophila* and vertebrates[24, 25]. In *C. elegans*, BMP/*sma* signaling regulates body size and male tail structure during development[14], but is not required for viability, which has greatly facilitated the discovery of its post-developmental functions in innate immunity[26], reproductive aging[27, 28] and chemosensation[29, 30]. We now reveal its crucial functions in regulating lipid metabolism. Such diverse effects of BMP signaling likely rely on the cooperation between transcription co-factors and Smads to regulate specific target genes. For example, BMP signaling controls the gene expression of cuticle collagen[31] and hedgehog-related warthog ligands[32] to regulate body size, while antimicrobial peptides[32–34] for immune responses. Through systematic analyses of metabolic gene expression profiles, we found that the mitochondrial targets of BMP signaling specifically mediate its effects on lipid accumulation but not on body size. Our results thus show that the

regulations of developmental growth and lipid metabolism by BMP signaling not only act in distinct somatic tissues but also have different molecular bases. Therefore, there might be tissue-specific transcription co-factors to accommodate the specificity and complexity of BMP signaling.

Cellular lipid metabolism is actively balanced between lipid synthesis and mobilization. The level of lipid storage is just a snapshot of this dynamic process. Previously, we coupled deuterium labeling with SRS microscopy for monitoring temporal dynamics of specific lipid molecules at the single lipid droplet level in living cells and organisms[7]. Here we demonstrated the application of this technique in tracking lipid influx and efflux to determine the rates of lipid synthesis and mobilization using live worms, which can also be widely applied to cell culture and other live organisms. Based on theses assays, we were able to associate the lipid phenotypes in the BMP signaling mutants with accelerated cellular lipid mobilization. This acceleration is accompanied with the transcriptional induction of several acyl-CoA synthetases (*acs-2, 11,15,16,17,18*) and carnitine palmitoyl transferases (*cpt-3, 4*), which catalyze essential steps of mitochondrial ß-oxidation that is the major catabolic process to consume fatty acids. Paradoxically, we also found that two

acyl-CoA dehydrogenases (*acdh-1, 2*) and one enoyl-CoA hydratase (*ech-6*) for short-chain fatty acids are transcriptionally down-regulated in the BMP mutants. These three genes are also down-regulated in fasted animals[35], suggesting that the oxidation of short-chain fatty acids might be inhibited upon induced lipid mobilization. Consistent with this idea, we detected the induction of several genes (*acox-5, F59F4.1, ech-8,* and *ech-9*) involved in peroxisomal β-oxidation, a catabolic process specifically for the oxidation of long-chain and very-long-chain fatty acids[36]. Furthermore, three fatty acid binding proteins (*lbp-1,5,7*) are also induced in the BMP mutants. This class of proteins functions as lipid chaperones to carry free fatty acids and their derivatives and to facilitate their trafficking between cellular organelles[37, 38], and their induction here might be responsible for transporting free fatty acids to mitochondria and peroxisomes for β-oxidation.

In addition to the transcriptional control of metabolic genes by BMP signaling, we discovered its previously unknown function in regulating mitochondrial dynamics. In the BMP mutants, mitochondrial architecture displays a more tubular, connected morphology. In a normal cell, mitochondria form a highly dynamic, interconnected network, and maintaining the architecture of this complex network requires active mitochondrial fission and fusion mediated by several large GTPases[18, 19]. Mitofusin is the GTPase essential for mitochondrial fusion, and the mutation of its *C. elegans* homolog *fzo-1* leads to mitochondrial fragmentation[20]. We found that the *fzo-1* mutant can rescue lipid accumulation in the BMP mutants, but does not affect body size. Therefore, remodeling of mitochondrial architecture specifically attributes to the lipid metabolic regulation by BMP signaling. Mitochondrial dynamics has been associated with the metabolic state of the cell[39], and recent studies show that tubular, connected mitochondrial architecture is necessary for fatty acid exchange during mitochondrial β-oxidation[40]. Our findings demonstrate BMP signaling as a crucial molecular link to coordinate mitochondrial dynamics and lipid catabolism. Considering high conservation of the BMP signaling pathway across species, similar regulatory mechanisms are expected to function in other systems. Previous studies showed that the larval transparency phenotype of the *Drosophila gbb*/BMP mutant is associated with decreased lipid stores in the larval fat body[41]. Future studies can confirm whether alterations in mitochondrial dynamics and gene expression are responsible for this developmental lipid phenotype, and whether BMP signaling coordinates mitochondrial homeostasis and lipid metabolism in adult flies and in mammalian systems. More importantly, further characterization of the mechanistic link between mitochondrial dynamics and BMP signaling can advance our understanding of this essential developmental pathway in metabolic regulations.

## Methods

**C. elegans strains and maintenance**. *C. elegans* strains were grown on standard NGM (nematode growth medium) plates with *E. coli* OP50 at 20 °C using standard protocols[42]. The following strains were used for this study: *raxIs21[rab-3p::mEosFP]*, N2, *sma-2(e502), sma-3(e491), sma-4(e729), sma-6(wk7), sma-9 (wk55), dbl-1(wk70), daf-4(e1364), daf-1(m40), daf-3(mgDf90), daf-5(e1386), daf-7 (e1372), daf-8(e1393), daf-14(m77), sma-1(e30), sma-3(wk30), sma-3(wk30); him-5 (e1490); qcEx24[sma-3p::GFP::sma-3 + rol-6], sma-3(wk30); him-5(e1490); qcEx52 [myo-2p::GFP::sma-3 + rol-6], sma-3(wk30); him-5(e1490); qcEx53[vha-6p::GFP:: sma-3 + rol-6], sma-3(wk30); him-5(e1490); qcEx5[dpy-7p::GFP::sma-3 + rol-6], acs-2(ok2457), fzo-1(tm1133), zcIs17[ges-1p::mitoGFP], sma-2(e502); zcIs17[ges-1p:: mitoGFP], sma-3(e491); zcIs17[ges-1p::mitoGFP], sma-3(wk30); zcIs17[ges-1p:: mitoGFP], sma-3(wk30); qcEx52[myo-2p::GFP::sma-3 + rol-6]; zcIs17[ges-1p:: mitoGFP], sma-3(wk30); qcEx5[dpy-7p::GFP::sma-3 + rol-6]; zcIs17[ges-1p:: mitoGFP], sma-3(wk30); qcEx53[vha-6p::GFP::sma-3 + rol-6]; zcIs17[ges-1p:: mitoGFP];* and *sma-2(e502); fzo-1(tm1133)*.

**Imaging chamber**. Multi-well plastic grid was printed using a 3D printer (Supplementary Fig. 1), then glued onto a large size (76 mm×83 mm) coverslip (Brain Research Laboratories, Newton, MA) using silicone gel.

**SRS microscopy and photo-highlighting**. The experimental set-up was built on an inverted microscope (IX81, Olympus, Shinjuku, Japan) as shown in Supplementary Fig. 2a, b. For SRS microscopy, spatially and temporally overlapped pulsed Pump (tunable from 720 to 990 nm, 7 ps, 80 MHz repetition rate) and Stokes (1064 nm, 5 ~ 6 ps, 80 MHz repetition rate, modulated at 8 MHz) beams provided by picoEMERALD (Applied Physics & Electronics, Berlin, Germany) were coupled into an inverted laser-scanning microscope (FV1000 MPE; Olympus) optimized for near-IR throughput. A 20 × air objective (UPlanSAPO; 0.75N.A.; Olympus) was used for imaging. A 60 × water objective (UPlanAPO/IR; 1.2N.A.; Olympus) was used for C-D signal and high magnification imaging. After passing through the sample, the forward going Pump and Stokes beams were collected in transmission by a water condenser. A high OD bandpass filter (890/220, Chroma, Bellows Falls, VT) was used to block the Stokes beam completely and to transmit only the Pump beam onto a large area Si photodiode for the detection of the stimulated Raman loss signal. The output current from the photodiode was terminated, filtered, and demodulated by a lock-in amplifier (HF2LI; Zurich Instruments, Zurich, Switzerland) at 8 MHz to ensure shot noise-limited detection sensitivity. $CH_2$ signals were imaged at 2845 cm$^{-1}$, $CD_2$ signals were imaged at 2110 cm$^{-1}$, and $CH_3$ signals were imaged at 2940 cm$^{-1}$. The same microscope has three continuous visible lasers (405, 488, and 559 nm) controlled by an Acoustic Optical Tunable Filter (AOTF) (Supplementary Fig. 2b). The 405 nm laser is used for photo-highlighting. The microscope is controlled by Olympus Fluoview 1000 software.

**EMS screening**. The *C. elegans* strain *raxIs21[rab-3p::mEosFP]* at the L4 stage were mutagenized with 0.05 M EMS (methanesulfonic acid, ethyl ester, Sigma, St. Louis, MO) at 20 °C for 4 h. After recovery, 120 healthy worms were hand picked as $P_0$ onto 10 standard NGM plates seeded with OP50. $P_0$ worms were transferred onto new OP50 plates everyday for 3 days. Gravid $F_1$ adults were bleached, and $F_2$ worms were synchronized at the L1 stage in M9 buffer overnight and transferred onto standard NGM plates seeded with OP50.

When they reached the first day of adulthood (one day after the late L4 larval stage, with fertilized eggs in the uterus), $F_2$ worms were washed-off from the plates and loaded into the imaging chamber at the density of about 120 worms per well. ×20 air objective (UPlanSAPO; 0.75N.A.; Olympus) and water condenser (IX2-TLW; 0.9N.A.; Olympus) were used for imaging. The worms in each well of the imaging chamber were scanned rapidly to examine lipid phenotypes using SRS microscopy (~1 s per worm). We screened for overall changes in lipid content (overall increased or decreased) and also for altered lipid distribution between three major lipid storage tissues of *C. elegans*: intestine, hypodermis, and oocytes. The criteria for altered lipid distribution were: (1) overall increase of lipid content levels in both somatic tissues (intestine and hypodermis), (2) overall decrease of lipid content levels in both somatic tissues; (3) enlarged lipid droplets in intestinal cells; (4) lipid content levels in oocytes higher than those in the intestine; (5) lipid content levels in the hypodermis higher than those in the intestine. At the same time, any mutant candidates with notable developmental delay were recorded.

Once the candidates with altered lipid distribution were identified, the head region of the worm was circled and photo-illuminated using the 405 nm laser for 15–30 s. After imaging all the worms, the chamber was transferred to a fluorescence stereoscope (SMZ1500, Nikon, Chiyoda, Japan), and the photo-highlighted candidates were picked using a mouth pipette.

**Candidate validation**. To confirm their lipid phenotypes, candidates were expanded into a population and imaged using SRS microscopy three times independently (by mounting ~20 worms onto a 2% agarose pad on a glass slide with 0.5% sodium azide and imaging them[43]). For these experiments, in addition to L1 synchronization, the EMS screen candidates and their controls were synchronized at L4 stage and imaged 12–18 h later after initiation of egg laying to ensure imaging of 1-day-old adult worms and eliminate any effect that might be caused by developmental delay. $CH_3$ signals derived from proteins were imaged by SRS at 2940 cm$^{-1}$, which show that total protein levels are not changed between wild type and mutant worms (Supplementary Fig. 7).

To quantify lipid levels in the intestine and hypodermis, the same parts of worm intestine and hypodermis were selected using the polygon selection tool in ImageJ (NIH) and the average pixel intensity was measured. The mutants were then normalized to their wild-type controls and relative SRS intensity changes were represented in the results. To image lipid distribution in the whole body, worms were z-sectioning scanned. A projection image was generated by "Maximum Intensity Z-projection" in ImageJ, and the average whole-body signaling intensity was compared between the mutants and their wild-type controls.

Upon validation, 57 mutants (*rax2 ~ rax58*) were reported as candidates with altered lipid phenotypes, and 77 false positive mutants were eliminated from further characterization.

**SNP mapping and sequencing**. The isolated *rax3*, *rax5*, *rax7*, and *rax10* mutants were mapped using rapid single-nucleotide polymorphism (SNP) mapping method[12], and sequenced using Nextseq 500 (Illumina, San Diego, CA). Point mutations were identified from analyses of sequencing results[13]. SNP mapping and next-generation sequencing characterized *rax5* as a *sma-2* mutant, *rax7* as a *sma-3* mutant, and *rax3* and *rax10* as *sma-4* mutants.

**Oil Red O staining**. Synchronized day-1 adult worms were collected, washed 3 times with M9 buffer, fixed in 50% isopropanol for 15 min and stained by 60% Oil Red O (Sigma) overnight at room-temperature[44]. 20–30 stained worms were mounted on a glass slide then imaged under a DIC microscope with a ×20 objective and a color camera (Zeiss AxioImager.M1, Oberkochen, Germany).

**Deuterated fatty acid supplementation**. OP50 bacterial culture was mixed well with 1 mM deuterated oleic acid (OA-$D_{34}$, Sigma), and then seeded onto NGM plates. To analyze the rate of lipid synthesis/incorporation, 1-day-old adult worms were fed with OA-$D_{34}$ supplemented bacteria and imaged using SRS microscopy at indicated time intervals. To analyze the rate of lipid mobilization, OA-$D_{34}$ labeled worms were transferred onto fresh NGM plates with unlabeled OP50 and imaged using SRS microscopy at indicated time intervals. Different from $CH_2$ signal imaging, $CD_2$ signals were imaged at 2110 $cm^{-1}$ and a ×60 water objective (UPlanAPO/IR; 1.2N.A.; Olympus) was used. The images were quantified in ImageJ. In each worm, the anterior first 2 pairs of cells were selected using the polygonal selection tool and the average pixel intensity was measured. The mutants were then normalized to their wild-type controls in the first time point and relative SRS intensity changes were represented in bar charts.

**qPCR**. Total RNA were isolated from about 1000 age-synchronized day-1 adult worms using Trizol extraction with column purification (Qiagen, Hilden, Germany). Synthesis of cDNA was performed using the amfiRivert Platinum cDNA Synthesis Master Mix (GenDEPOT, Barker, TX). Quantitative PCR was performed using Kapa SYBR fast qPCR kit (Kapa Biosystems, Wilmington, MA) in a 96-well Eppendorf Realplex 4 PCR machine (Eppendorf, Hamburg, Germany). Values were normalized to *act-1* and *rpl-32* as internal controls. All data shown represent three biologically independent samples. Statistical analysis was performed using the paired two-sample *t*-test for pairwise comparisons. Quantitative PCR (qPCR) primer sequences are listed in Supplementary Table 1.

**RNA-seq**. Three repeats of *sma-4(rax3)*, *sma-2(rax5)* and out-crossed wild-type sibling worms were synchronized and grown to day-1 adults. Total RNA were isolated from about 2000 worms using Trizol extraction with column purification (Qiagen). Sequencing libraries were prepared using the TruSeq Stranded mRNA Sample Preparation kit (Illumina) according to manufacturer's instructions. Libraries were pooled together and sequenced using Illumina NextSeq 500 system. Sequencing reads were aligned to *C. elegans* WS235 genome using Tophat2[45]. HTseq-count[46] was used to count reads mapped to each gene and counting data was imported to EdgeR[47] for statistical analysis. Statistical significance was defined by adjusted *P* value (false discovery rate, FDR) of <0.05.

**Mitochondria morphology analysis**. To assess mitochondrial morphology in the *sma* mutants and *sma-3* rescue stains, *sma-2(e502)*, *sma-3(e491)*, *sma-3(wk30); him-5(e1490)*; *qcEx52[myo-2p::GFP::sma-3 + rol-6]*, *sma-3(wk30); him-5(e1490); qcEx5[dpy-7p::GFP::sma-3 + rol-6]* and *sma-3(wk30); him-5(e1490);* and *qcEx53 [vha-6p::GFP::sma-3 + rol-6]* were crossed to transgenic worms carrying the array *zcIs17[ges-1p::mitoGFP]*. 1-day-old adult transgenic worms were imaged using confocal fluorescence microscopy. At least 40 animals from each genotype were analyzed each time and the analysis was repeated 3 times. The 488 nm laser, ×60 oil objective (PlanAPO N, 1.42N.A., Olympus) and additional digital 3X zoom were used for imaging set-up. The confocal fluorescence microscope was controlled by Olympus Fluoview 1000 software. For each genotype, the anterior part of the intestine (first 3-4 pairs of intestinal cells) was imaged. The obtained images were pseudo-numbered and analyzed in a double-blinded manner by grouping them into tubular, intermediate and fragmented morphology as shown by representative images.

**Statistical analyses**. Data are expressed as mean ± s.d. unless otherwise indicated. Statistical analyses for all data were performed by Student's *t*-test (unpaired, two-tailed) with at least 3 replicates, unless otherwise indicated. Statistical analyses were performed in Excel (Microsoft) or GraphPad Prism (GraphPad Software Inc., La Jolla, CA, USA).

**Data availability**. RNA-seq data that support the findings of this study have been deposited in the NCBI Sequence Read Archive (SRA, http://www.ncbi.nlm.nih.gov/sra) under the BioProject ID PRJNA395096. The accession codes for each of the biological samples are SAMN07370969, SAMN07370970, SAMN07370971, SAMN07370972, SAMN07370973, SAMN07370974, SAMN07370975, SAMN07370976, and SAMN07370977. All other data supporting the findings of this study are available from the corresponding authors upon request.

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

## Acknowledgements

We thank C. Savage-Dunn for *sma-3* rescue strains, and K. Sheng and C. Zong for the help of next-generation sequencing and analysis. This work was supported by NIH grants R01AG045183 (M.C.W.), R01AT009050 (M.C.W.), DP1DK113644 (M.C.W.), R21EB022302 (M.C.W.), and by HHMI faculty scholar (M.C.W.), HHMI pre-doctoral fellowships (A.S.M.), and NSF Graduate Research Fellowship (H.L.).

## Author contributions

Y.Y. and M.C.W. conceived the study; Y.Y. and M.C.W. designed the experiments; Y.Y., A.S.M., H.L., and M.C.W. performed the experiments; Y.Y., A.S.M., and M.C.W. wrote the manuscript.

## Additional information

**Competing interests:** The authors declare no competing financial interests.

