## [Peer Review File · Nature Communications]

Reviewers' comments:

Reviewer #1 (Remarks to the Author):

In this manuscript, Meng Wang and colleagues develop and apply a novel method to quickly tag mutants in a screen, allowing for rapid isolation. Application of this new approach in combination with state-of-the-art SRS microscopy then allowed them to identify a new class of mutants affecting lipid metabolism in *C.elegans*.

The data presented here are strong, the conclusions well justified and the paper is very well written. There have been occasional sideline observations that BMPs might be important for regulation of metabolism, but this unbiased identification of the genes in *C.elegans* puts them squarely on the map, which in my assessment is a very significant discovery.

An interesting point about the paper that the authors show, but don't discuss much is that apparently rescue of expression in the pharyngeal muscle was sufficient to rescue the phenotype. This leads me to wonder whether there are any developmental phenotypes in this tissue. One could imagine that some of the mitochondrial expression changes are due to a change in food intake (which the authors to address in a sentence as not changed, would be good to show). It's also possible that this tissue has a specific and larger need for energy generation in mitochondria. I would encourage the authors to show more of these data and add a short discussion of this point.

Otherwise, this is a very strong paper, combining innovative methodology with novel biological insight and thus should be a strong candidate for Nature Communications.

Reviewer #2 (Remarks to the Author):

This manuscript describes the usage of the photo-convertible fluorescent protein mEosFP in genetic screens to photo-highlight candidate mutant worms. Using this method, the authors performed a large scale genetic screen for genes involved in regulating lipid distribution via a previously described method using SRS (stimulated Raman scattering) microscopy. In particular, the authors found that mutations in the BMP pathway cause reduced lipid accumulation in somatic tissues. Using tissue specific rescue experiments, the authors showed that BMP signaling regulates lipid metabolism independent of its well established role in body size regulation. They further showed that BMP signaling regulates lipid mobilization by controlling mitochondria beta-oxidation.

The most significant part of this manuscript is the development of the photo-highlighting method. This method allows for effective recovery of the desired mutants in large scale genetic screens under the microscope, and will be very useful to the *C. elegans* community as well as researchers working with optically transparent organisms. This method, however, will only be useful to researchers aiming to identify viable and fertile mutants from a non-clonal screen. Nevertheless, this is a really neat method.

The other important finding of the manuscript is the role of BMP signaling in lipid metabolism. I do have a number of questions/comments regarding this part of the manuscript.

1) The senior author has previously published a paper (Wang et al., 2011, Nature Methods) describing that *sma-6(RNAi)* caused increased lipid accumulation using the same SRS-based method. However, in the current manuscript, they report that there is decreased lipid accumulation in *sma-6* mutant as well as other BMP pathway mutants. These discrepancies raised significant concerns on the reproducibility of the SRS-based screening method for lipid regulators.

2) The authors described that day 1 adults at the F2 generation were screened using SRS microscopy. Have the authors taken into account variations of the developmental rates of various mutants? For example, mutations in the BMP pathway are known to develop slower than wild-type animals. Are there any variations of lipid accumulation in worms at different developmental stages? Along the same line, Figure 2C showed representative images of different classes of mutants isolated from the screen. Are all mutants shown at exactly the same developmental stage?

3) Regarding the site of action of BMP signaling in regulating lipid metabolism, the authors commented that expressing *sma-3* in pharyngeal muscles can fully rescue the lipid accumulation phenotype of *sma-3(wk30)* mutants. In the same figure (figure 3e), intestinal expression of *sma-3* also partially rescued the lipid phenotype. The authors should comment on the role of *sma-3* in the pharynx and in the intestine. Representative SRS images of these rescue experiments should be shown.

4) The sample sizes for the iSRS experiments are rather low (n=5).

5) In figure 5d, what stage worms were used to analyze mitochondria morphology? The photos are of low magnification and hard to evaluate. How was the quantification done, using the same cells in worms of different genotypes or in all intestinal cells of worms of different genotypes? Can expression of *sma-3* (and presumably other BMP pathway components) in pharyngeal muscles rescue the INTestinal mitochondria morphology defects? If not, how do the authors explain the rescue of the lipid accumulation phenotype of *sma-3* mutants by pharyngeal muscle specific expression of *sma-3*?

Reviewer #3 (Remarks to the Author):

In this article, Yu et al. applied stimulated Raman scattering (SRS) microscopy to screen lipid phenotypes of *C. elegans* after EMS mutagenesis, and then employed photoconversion of a fluorescent protein to mark the positive mutants thus facilitate sorting. By the combination of these techniques, the authors identified 57 mutations that exhibit the alteration in lipid distribution. They picked 4 from them that were later identified be with BMP signaling pathway. By characterization more known BMP-associated mutants, they revealed the link between BMP and mitochondrial lipid metabolism. The authors claim that the combination of SRS and photo-highlighting technique improves the screening process dramatically and that they have found a novel role of BMP in lipid metabolism.

However, I have serious problems about the technical validity. There is a fundamental mismatch between high throughput screening and nonlinear optical microscopy (particularly single-color SRS imaging of lipid) on large organisms like worms.

Major problems:

SRS is a nonlinear optical sectioning technique. Each SRS image can only measure lipid distribution from a thin z layer of about a few micron. In contrast, a worm can be 50 micron or thicker. Hence, one cannot draw quantitative conclusion based on a single SRS image. Yet, the method reported is trying high-throughput screen with such an optical sectioning technique. The only technically valid approach is to perform a z-stack on each worm. But then the speed would go down drastically. There is a fundamental mismatch between high throughput screening and nonlinear optical microscopy on large organisms.

The authors only took a single 2845 cm⁻¹ image for all of their animals, which is very problematic. As it is well known to Raman community, the lipid channel and the protein channel have mutual cross-talk in Raman spectra. One has to take at least two images (one 2845 cm⁻¹ and one 2940 cm⁻¹) to measure both lipid and protein components, and do a spectral unmixing. Without such a two-channel unmixing, the 2845 cm⁻¹ channel is very likely to be contaminated by the protein channel.

For screening lipid phenotypes, what are the criteria for 'altered lipid distribution'? How could the authors take into account the variation between individuals as it seems the phenotype identification and classification are based only on one worm for each mutant. Heterogeneity must exist among the wild type and among each mutant. In addition to the qualitative classification in Fig. 2, the authors are advised to quantitatively verify the lipid content and LDs in more worms in those 57 mutants.

Among those 57 mutants that exhibit altered phenotypes, the authors have only mentioned the genotype for only 4 of them among mutant class I. How did the authors select these 4 out of 57 mutants? The authors are expected to show the genetic change of the other mutants in class I as well as in other classes, which will be informative to the field.

By SNP mapping, the authors attribute the lipid alteration to mutations in genes involved in BMP signaling pathway. But by EMS mutagenesis, mutation happens at a frequency of one induced mutation every 400,000 nucleotides per mutagenized genome (Genome Res. 2007 May; 17(5): 649–658.). So, it's highly possible that each selected mutant bears multiple mutations. How could the authors rule out the contribution of mutations in other genes?

Given that the important role of BMP signaling in lipid metabolism and mitochondrial homeostasis has already been shown in flies and mice (Antioxid Redox Signal. 2013 Jul 20; 19(3): 243–257.; Dev Biol. 2010 Jan 15; 337(2): 375–385.), the novelty of biological discover in this article is lacking.

By feeding worms isotope labeled oleic acid and corresponding pulse-chase experiment, the authors suggest the mutants have unchanged lipid uptake but elevated mobilization that leads to decreased lipid distribution. However, this doesn't eliminate the possibility of altered lipid de novo synthesis as this might also be affected by BMP signaling pathway. In figure 5 a, those genes responsible for de novo synthesis are indeed up-regulated.

Other Problems:

The authors claim 'photo-highlighting' strategy improves screening and sorting speed by almost 100 times compared to traditional methods. References should be added for the estimation of traditional speed.

SRS microscopy targeting CH₂ bonds here is firstly used to quantify lipid distribution in *C. elegans*. But SRS signal intensity is prone to the bias of focus position. As the authors solely rely on SRS intensity at 2845cm⁻¹, errors could be introduced if not all worms were imaged at the same depth, or even if worms were different in size since the diameter of the worm is usually larger than the axial point spread function.

In the 'photo-highlighting' process, worms bearing phenotypes of interested are identified and then marked by the conversion of mEOS. Is the photobleaching of converted mEOS by SRS laser severe? What are the high/low laser powers used in Fig.2?

The sample size is too small (only 5 worms) and the difference is not so significant for figure4 c and d, on which the whole story of increased lipid mobilization is based. The authors should also elaborate how the quantification here was done, as it seems that SRS intensity was measured in the whole worm that may contain organs such as ovaries that are rich in lipid in a certain life stage.

And what's interesting is how the single amino acid mutation (rax3, rax5, rax10) will cause a dramatic change in lipid distribution, which is completely missing in the article.

Reviewer #1

In this manuscript, Meng Wang and colleagues develop and apply a novel method to quickly tag mutants in a screen, allowing for rapid isolation. Application of this new approach in combination with state-of-the-art SRS microscopy then allowed them to identify a new class of mutants affecting lipid metabolism in C.elegans.

The data presented here are strong, the conclusions well justified and the paper is very well written. There have been occasional sideline observations that BMPs might be important for regulation of metabolism, but this unbiased identification of the genes in C. elegans puts them squarely on the map, which in my assessment is a very significant discovery.

An interesting point about the paper that the authors show, but don't discuss much is that apparently rescue of expression in the pharyngeal muscle was sufficient to rescue the phenotype. This leads me to wonder whether there are any developmental phenotypes in this tissue. One could imagine that some of the mitochondrial expression changes are due to a change in food intake (which the authors to address in a sentence as not changed, would be good to show). It's also possible that this tissue has a specific and larger need for energy generation in mitochondria. I would encourage the authors to show more of these data and add a short discussion of this point.

Otherwise, this is a very strong paper, combining innovative methodology with novel biological insight and thus should be a strong candidate for Nature Communications.

Response: We really appreciate that the reviewer finds our work with high novelty and significance. We also thank the reviewer's comment regarding the effect of BMP signaling on the development and function of the pharyngeal muscle and its association with mitochondrial phenotypes. Based on morphological observation, the structure of the pharynx is not altered in the BMP signaling mutants. Moreover, functional characterization of pharyngeal pumping shows that the BMP signaling mutants have similar rates as wild type controls (new Supplementary Figure 4a). Therefore, although the regulatory effect of BMP signaling on lipid metabolism is mediated by its activity in the pharyngeal muscle, this is not due to alterations in pharyngeal development or its physical activities.

We agree with reviewers 1 and 2 that it is interesting to examine the tissue-specificity of BMP signaling in regulating mitochondrial dynamics. We have conducted these experiments and found that the restoration of *sma-3* expression in the pharyngeal muscle but not in the hypodermis suppresses increased mitochondrial tubular morphology in the intestine of the *sma-3* mutant (new Figure 6b). This tissue-specificity is consistent with the fat storage regulation by *sma-3* (new Figure 3e). These new results further support our model that BMP signaling regulates fat storage via tuning mitochondrial dynamics, and reveal that BMP functions in the pharyngeal muscle to regulate intestinal mitochondrial dynamics and fat storage cell non-autonomously. As suggested by the reviewer, we have included these new data and discussion in the revised manuscript. We would like to thank reviewers 1 and 2 for their constructive suggestions.

Reviewer #2

This manuscript describes the usage of the photo-convertible fluorescent protein mEosFP in genetic screens to photo-highlight candidate mutant worms. Using this method, the authors performed a large scale genetic screen for genes involved in regulating lipid distribution via a previously described method using SRS (stimulated Raman scattering) microscopy. In particular, the authors found that mutations in the BMP pathway cause reduced lipid accumulation in somatic tissues. Using tissue specific rescue experiments, the authors showed that BMP signaling regulates lipid metabolism independent of its well established role in body size regulation. They further showed that BMP signaling regulates lipid mobilization by controlling mitochondria beta-oxidation.

The most significant part of this manuscript is the development of the photo-highlighting method. This method allows for effective recovery of the desired mutants in large scale genetic screens under the microscope, and will be very useful to the C. elegans community as well as researchers working with optically transparent organisms. This method, however, will only be useful to researchers aiming to identify viable and fertile mutants from a non-clonal screen. Nevertheless, this is a really neat method.

The other important finding of the manuscript is the role of BMP signaling in lipid metabolism. I do have a number of questions/comments regarding this part of the manuscript.

Response: We thank the reviewer's comment that the development of the photo-highlighting method and the finding of BMP signaling in lipid metabolism regulation are both important.

1) The senior author has previously published a paper (Wang et al., 2011, Nature Methods) describing that sma-6(RNAi) caused increased lipid accumulation using the same SRS-based method. However, in the current manuscript, they report that there is decreased lipid accumulation in sma-6 mutant as well as other BMP pathway mutants. These discrepancies raised significant concerns on the reproducibility of the SRS-based screening method for lipid regulators.

Response: We really appreciate that the reviewer brought the phenotypic discrepancies between *sma-6* RNAi and mutation to our attention. Even though RNAi based screens in *C. elegans* has been very efficient and fruitful leading to new gene discoveries, there several concerns regarding RNAi knockdown, including off-target effects, partial/weak loss-of-function and heterogeneity of knockdown efficiency between different tissues. On the other hand, genetic mutations identified from EMS screens have well-defined target genes, mostly cause changes in protein structure and function, and are homogeneously efficient at the whole organism level. It is not rare that for the same gene, its RNAi knockdown and genetic mutation result in distinct phenotypes. In general, phenotypic results based on genetic mutations are more reliable. In particular, the *sma-6(wk7)* mutant that we used in this paper is a null allele (Krishna et al., Development 1999), which results in complete loss-of-function of *sma-6* at the whole organism level. *sma-6* RNAi knockdown on the other hand only leads to partial loss-of-function in some tissues (e.g. pharyngeal muscle is insensitive to RNAi knockdown). Therefore, the phenotypic discrepancy of *sma-6* is related to the problem associated with its RNAi knockdown, but not to the reproducibility of the SRS-based screening method.

In both RNAi and EMS screens, the phenotypes are reproducible, which proves the robustness of the SRS-based screening method. The identification of multiple components in the same

BMP signaling pathway through one EMS screen further demonstrate the reliability of SRS-based EMS screens, which highlights the importance to develop this new method for discovering lipid regulators.

2) The authors described that day 1 adults at the F₂ generation were screened using SRS microscopy. Have the authors taken into account variations of the developmental rates of various mutants? For example, mutations in the BMP pathway are known to develop slower than wild-type animals. Are there any variations of lipid accumulation in worms at different developmental stages? Along the same line, Figure 2C showed representative images of different classes of mutants isolated from the screen. Are all mutants shown at exactly the same developmental stage?

Response: We appreciate the reviewer's comment regarding the effect of developmental timing on lipid phenotypes. For EMS mutagenesis screen, F₂ worms were synchronized at the L1 stage in M9 buffer overnight and transferred onto standard NGM plates seeded with OP50 bacteria (standard laboratory conditions for growing worms). Then, 1-day-old adult worms (one day after the late L4 larval stage, with fertilized eggs in their uterus) were imaged by SRS microscopy. With this design, most of the worms were at the similar stage when imaging. During the screening process, any worms with severe developmental delay were noted. In fact, more than 100 mutants were identified with dauer arrested phenotypes, which are reported in Figure 2b.

Importantly, during the verification steps following the screen, the mutants were always imaged at the same stage. Specifically, the mutants and their controls shown in Figure 2c are day-1-old adults, which were synchronized at the L4 stage and imaged 12-18 hours later after initiation of egg laying. During the reproductive period at adulthood, fat storage levels do not alter much within the 12-hours time window. We are sorry that we did not clearly state these experimental designs in detail, and have revised the method section thoroughly.

*3) Regarding the site of action of BMP signaling in regulating lipid metabolism, the authors commented that expressing *sma-3* in pharyngeal muscles can fully rescue the lipid accumulation phenotype of *sma-3(wk30)* mutants. In the same figure (figure 3e), intestinal expression of *sma-3* also partially rescued the lipid phenotype. The authors should comment on the role of *sma-3* in the pharynx and in the intestine. Representative SRS images of these rescue experiments should be shown.*

Response: We thank the reviewer for the suggestions. In the revised manuscript, we have included the representative SRS images in new Supplementary Figure 5, and we have also discussed the roles of *sma-3* in the pharynx and in the intestine.

"Pharynx and intestine, together with the rectum, form the digestive tract of *C. elegans*. Even though pharynx has been thought as an organ solely for the grinding and transportation of the bacteria to the intestine, the anatomy of the pharynx suggests that it could be an active signaling hub as well. First, in the posterior, six valve cells connect the last pharyngeal muscular cell, to the first pair of intestinal cells. Thus, the pharynx can be considered as a continuum of the anterior intestine. Second, the basal surface of the pharynx is lined by basal lamina, which is in contact with the pseudocoelomic cavity. Therefore, BMP signaling in the pharynx might either directly communicate with the intestine via the connecting valve cells or there may be indirect

endocrine signaling between the pharynx and rest of the body including the intestine via the pseudocoelom.”

4) The sample sizes for the iSRS experiments are rather low (n=5).

Response: We have increased the sample size in new experiments (n=16, new Figure 4), and the previous results are now shown in the new Supplementary Figure 6.

5) In figure 5d, what stage worms were used to analyze mitochondria morphology? The photos are of low magnification and hard to evaluate. How was the quantification done, using the same cells in worms of different genotypes or in all intestinal cells of worms of different genotypes? Can expression of sma-3 (and presumably other BMP pathway components) in pharyngeal muscles rescue the INTESTINAL mitochondria morphology defects? If not, how do the authors explain the rescue of the lipid accumulation phenotype of sma-3 mutants by pharyngeal muscle specific expression of sma-3?

Response: For mitochondrial morphology analysis, day-1-old adult worms that express GFP with mitochondrial localization signal (mitoGFP) in the intestine (*[Pges-1::mitoGFP]*) were imaged using confocal fluorescence microscopy. At least 40 animals from each genotype were analyzed each time and the analysis was repeated 3 times. The confocal fluorescence microscope has three continuous visible lasers (405 nm, 488 nm and 559 nm) controlled by an Acoustic Optical Tunable Filter (AOTF). The 488 nm laser, a 60x oil objective (PlanAPO N, 1.42 N.A., Olympus), and a digital zoom-in (3X) were used. The confocal fluorescence microscope was controlled by Olympus Fluoview 1000 software. For each genotype, the anterior part of the intestine (first 3-4 pairs of intestinal cells) was imaged. The obtained images were pseudo-numbered and analyzed in a double-blinded manner by grouping them into tubular, intermediate or fragmented morphology as shown by representative images. We have revised the method section to explain mitochondrial morphology analysis in detail.

We agree with reviewers 1 and 2 that it is interesting to examine the tissue-specificity of BMP signaling in regulating mitochondrial dynamics. We have conducted these experiments and found that the restoration of *sma-3* expression in the pharyngeal muscle but not in the hypodermis suppresses increased mitochondrial tubular morphology in the intestine of the *sma-3* mutant (new Figure 6b). This tissue-specificity is consistent with the fat storage regulation by *sma-3* (new Figure 3e). These new results further support our model that BMP signaling regulates fat storage via tuning mitochondrial dynamics, and reveal that BMP functions in the pharyngeal muscle to regulate intestinal mitochondrial dynamics and fat storage cell non-autonomously. As suggested by the reviewer, we have included these new data and discussion in the revised manuscript. We would like to thank reviewers 1 and 2 for their constructive suggestions.

Reviewer #3:

In this article, Yu et al. applied stimulated Raman scattering (SRS) microscopy to screen lipid phenotypes of C. elegans after EMS mutagenesis, and then employed photoconversion of a fluorescent protein to mark the positive mutants thus facilitate sorting. By the combination of these techniques, the authors identified 57 mutations that exhibit the alteration in lipid distribution. They picked 4 from them that were later identified be with BMP signaling pathway. By characterization more known BMP-associated mutants, they revealed the link between BMP

and mitochondrial lipid metabolism. The authors claim that the combination of SRS and photo-highlighting technique improves the screening process dramatically and that they have found a novel role of BMP in lipid metabolism.

However, I have serious problems about the technical validity. There is a fundamental mismatch between high throughput screening and nonlinear optical microscopy (particularly single-color SRS imaging of lipid) on large organisms like worms.

Response: Although we truly value the reviewer's technical concerns regarding the application of single-color SRS imaging for high-throughput screening and the requirement of spectral unmixing for SRS-based lipid quantification, these technical requirements are not crucial for high-throughput genetic screens in biological systems or for relative quantitation between genotypes in live subjects. Detailed discussions are the following.

Major problems:

SRS is a nonlinear optical sectioning technique. Each SRS image can only measure lipid distribution from a thin z layer of about a few micron. In contrast, a worm can be 50 micron or thicker. Hence, one cannot draw quantitative conclusion based on a single SRS image. Yet, the method reported is trying high-throughput screen with such an optical sectioning technique. The only technically valid approach is to perform a z-stack on each worm. But then the speed would go down drastically. There is a fundamental mismatch between high throughput screening and nonlinear optical microscopy on large organisms.

Response: The goal of large-scale genetic screens is to identify candidate mutants in a high-throughput and high-speed manner. Thus, it is a common strategy that a simplified method is employed to facilitate high-throughput genetic screens, and the identified candidates are further validated using more sophisticated and stringent methods (false positive candidates will be eliminated through these steps). In our studies, we have chosen one layer scanning in primary screens, which dramatically increases the screening efficiency. Following the screens, z-stack imaging was used to confirm final positive hits. We have also used alternative Oil Red O staining method to confirm the quantification results based on SRS imaging (new Supplementary Figure 3).

To deal with variable background signals, it might seem that there is a mismatch between high-throughput screening and nonlinear optical microscopy. With the simplicity of worm models, we have successfully combined them to conduct the first SRS-based high-throughput genetic screen, and identified novel BMP mutants with misregulations of lipid metabolism. Our studies thus demonstrated the feasibility of nonlinear optical microscopy in high-throughput screening. We would also like to emphasize that the newly developed photo-highlighting method is not limited to the SRS microscope and can be broadly applied to a variety of different microscopy systems.

The authors only took a single 2845 cm⁻¹ image for all of their animals, which is very problematic. As it is well known to Raman community, the lipid channel and the protein channel have mutual cross-talk in Raman spectra. One has to take at least two images (one 2845 cm⁻¹ and one 2940 cm⁻¹) to measure both lipid and protein components, and do a spectral unmixing.

Without such a two-channel unmixing, the 2845 cm⁻¹ channel is very likely to be contaminated by the protein channel.

Response: We agree with the reviewer that when quantifying absolute levels of lipid contents, spectral unmixing is required to subtract the contamination from proteins (a small portion 15% of 2845 cm⁻¹ signals are derived from proteins (Yu, Chen, et al. *Chem. Sci.*, 2012, 3, 2646)). But in our studies, we were comparing relative changes of lipid content levels between mutants and their controls. In this case, single-channel SRS imaging (2845 cm⁻¹) is sufficient for quantitative comparison, because total protein levels remain unchanged between mutants and their controls. We demonstrated that this quantification method gives similar results as TLC/GC (Wang et al, *Nature Methods* 2011) biochemical assays. In the revised manuscript, we have provided new supplementary information to show unchanged protein levels at 2940 cm⁻¹ (new Supplementary Figure 7). Moreover, when comparing the lipid phenotypes between mutants and their controls, we focused on the signals from lipid droplets, which contain only lipids.

For screening lipid phenotypes, what are the criteria for ‘altered lipid distribution’? How could the authors take into account the variation between individuals as it seems the phenotype identification and classification are based only on one worm for each mutant. Heterogeneity must exist among the wild type and among each mutant. In addition to the qualitative classification in Fig. 2, the authors are advised to quantitatively verify the lipid content and LDs in more worms in those 57 mutants.

Response: For screening lipid phenotypes, we examined lipid content in three major lipid storage tissues of *C. elegans*, including two somatic tissues, intestine and hypodermis, and oocytes. In wild type controls, lipid content levels in the intestine are much higher than those in the hypodermis or oocytes. The criteria for “altered lipid distribution” are: 1) overall increase of lipid content levels in both somatic tissues 2) overall decrease of lipid content levels in both somatic tissues; 3) enlarged lipid droplets in intestinal cells; 4) lipid content levels in oocytes higher than those in the intestine; 5) lipid content levels in the hypodermis higher than those in the intestine. These criteria have classified the identified mutants into five different phenotypic groups. To address the reviewer’s concern, we have revised the main text to highlight these criteria for textual clarity and the method section to explain these criteria.

The reviewer is right that heterogeneity exists among even genetically identical individuals. However, during an EMS genetic screen, each worm is genetically different from the others and worms with altered lipid phenotypes (mutants) are selected as candidates (including true positive hits and false positive hits). In the retests following the primary screen, each mutant will be expanded into a population, and multiple individuals will be examined to confirm the phenotype. We have followed these standard procedures in our studies. For each mutant isolated from the primary screen, 15-20 progenies of this mutant were imaged to confirm their phenotypes. Through these retests, 77 mutants were identified as false positives and eliminated from further characterization. For the reported 57 mutants in this manuscript, all of them were validated three times independently with at least 15 worms for each time. To address the reviewer’s concern, we have included more details in the method section.

We appreciate the reviewer’s suggestion to report quantification results of all 57 mutants. However this manuscript is to demonstrate the development of the photo-highlighting approach for microscopic screening, and the identification of 57 mutants and in-depth characterization of four BMP mutants among them has proven the success of this new method. We think the analyses of all 57 mutants are out of the scope of the current manuscript.

Among those 57 mutants that exhibit altered phenotypes, the authors have only mentioned the genotype for only 4 of them among mutant class I. How did the authors select these 4 out of 57 mutants? The authors are expected to show the genetic change of the other mutants in class I as well as in other classes, which will be informative to the field.

Response: The four mutants were selected based on their interesting low fat storage and small body size phenotypes, suggesting that they may function in the same genetic pathway. We appreciate the reviewer's interest in characterizing all 57 mutants. However this manuscript is to demonstrate the development of the photo-highlighting approach for microscopic screening, and the identification of 57 mutants and in-depth characterization of four BMP mutants among them has proven the success of this new method. We think the analyses of all 57 mutants are out of the scope of the current manuscript.

By SNP mapping, the authors attribute the lipid alteration to mutations in genes involved in BMP signaling pathway. But by EMS mutagenesis, mutation happens at a frequency of one induced mutation every 400,000 nucleotides per mutagenized genome (Genome Res. 2007 May; 17(5): 649–658.). So, it's highly possible that each selected mutant bears multiple mutations. How could the authors rule out the contribution of mutations in other genes?

Response: We appreciate the reviewer's concern about background mutations that occur in EMS mutagenesis. To eliminate the effects from secondary background mutations, we followed the routine procedures in the *C. elegans* genetics field (Davis et al, BMC genomics 2005; Sarin et al, Nature Methods 2008; Zuryn and Jarriault, Worm 2013). 1) After outcrossing into wild-type genetic background 8 times, secondary mutations that are not related to the phenotype were largely removed; 2) SNP mapping narrowed down genomic regions carrying the mutations responsible for the phenotype; 3) bioinformatic platform of deep-sequencing data analysis identified the mutations; 4) genetic rescue experiments confirmed that the identified mutations are the causal factors of the phenotype. Furthermore, for each gene that we identified, we have used at least two independent mutant alleles to validate their phenotypes, and therefore the likelihood that the background mutation(s) cause the phenotype is statistically impossible.

Given that the important role of BMP signaling in lipid metabolism and mitochondrial homeostasis has already been shown in flies and mice (Antioxid Redox Signal. 2013 Jul 20; 19(3): 243–257.; Dev Biol. 2010 Jan 15; 337(2): 375–385.), the novelty of biological discover in this article is lacking.

Response: We thank the reviewer bring these studies to our attention. As commented by Reviewer 1, "there have been occasional sideline observations that BMPs might be important for regulation of metabolism, but this unbiased identification of the genes in *C. elegans* puts them squarely on the map.", our work provides systematic and mechanistic characterization of the BMP signaling pathway in regulating lipid metabolism.

Previous literatures mainly focused on the role of BMP signaling during development (e.g. Dev Biol. 2010 Jan 15; 337(2): 375–385), and the mice study (Antioxid Redox Signal. 2013 Jul 20; 19(3): 243–257) characterized the cell-autonomous effect of BMP7 on mitochondrial respiration and fatty acid uptake in brown adipose tissues. These work do not comprise the novelty of our findings. Our work not only delineated the whole BMP signaling pathway in regulating lipid metabolism at adulthood, but also demonstrated that this regulation is mediated by a previously

unknown cell non-autonomous mechanism to control mitochondrial dynamics and beta-oxidation. Therefore, we believe our findings provide significant and new advances for understanding the biological functions of BMP signaling.

By feeding worms isotope labeled oleic acid and corresponding pulse-chase experiment, the authors suggest the mutants have unchanged lipid uptake but elevated mobilization that leads to decreased lipid distribution. However, this doesn't eliminate the possibility of altered lipid de novo synthesis as this might also be affected by BMP signaling pathway. In figure 5 a, those genes responsible for de novo synthesis are indeed up-regulated

Response: We appreciate the reviewer's insight. In our unpublished work, we have used deuterium-labeled glucose to measure de novo lipid synthesis in different biological systems. Based on these studies, we found that de novo lipid synthesis has very little contribution to fat storage in *C. elegans*. More importantly, transcriptome analysis of metabolic genes did not show that genes responsible for de novo synthesis are up-regulated. Figure 5a shows the up-regulation of mitochondrial β -oxidation and peroxisomal β -oxidation genes and unchanged expression of other lipid catabolic genes, such as lipolysis and TCA cycle genes. In the revised manuscript, we have included new data showing the unaltered expression levels of de novo fatty acid synthesis genes in the BMP mutants and their controls (new Figure 5a).

Other Problems:

The authors claim 'photo-highlighting' strategy improves screening and sorting speed by almost 100 times compared to traditional methods. References should be added for the estimation of traditional speed.

Response: We appreciate the reviewer's suggestion. In a traditional experimental set-up, we need to mount worm onto an agarose pad on a slide, image their phenotypes, find candidate, and recover candidates one by one from the slide. We have recorded the time for finishing one worm using the traditional method, and it is about 5 minutes. We now report this number in the revised manuscript. While using the photo-highlighting strategy, we can mount 1000~2000 worms in the imaging chamber and finish imaging and recovery within 30 minutes. Therefore, this new strategy costs about 100 times less time than the traditional method. Unfortunately, no reference is available reporting the time of the traditional method.

SRS microscopy targeting CH₂ bonds here is firstly used to quantify lipid distribution in c. elegance. But SRS signal intensity is prone to the bias of focus position. As the authors solely rely on SRS intensity at 2845cm⁻¹, errors could be introduced if not all worms were imaged at the same depth, or even if worms were different in size since the diameter of the worm is usually larger than the axial point spread function.

Response: In our previous studies, we have demonstrated that in the *C. elegans* system, SRS-based measurement at 2845cm⁻¹ is a very reliable way to quantify lipid content levels at the whole organism, and its results are very consistent with TLC/GC biochemical methods (Wang et al, *Nature Methods* 2011). During SRS imaging, each worm was imaged at the same depth, and the average signal intensity is used for quantitative comparison. Given that lipid droplets distribute quite evenly along Z-axis of the worm intestine, lipid content levels at different depth are similar. In addition, worms are always synchronized at day-1 adulthood for imaging, which

ensures their similar diameter size. Furthermore, we have also confirmed the SRS-based quantification results with other methods (new Supplementary Figure 3).

In the 'photo-highlighting' process, worms bearing phenotypes of interested are identified and then marked by the conversion of mEOS. Is the photobleaching of converted mEOS by SRS laser severe? What are the high/low laser powers used in Fig.2?

Response: In our screen condition, we did not observe that the converted mEosFP is photobleached by the SRS laser. In Fig. 2, the high laser power is 600 mW for pump/OPO (816 nm), 400 mW for Stokes/IR (1064 nm); the low laser power is 200 mW for pump/OPO (816 nm), 400 mW for Stokes/IR (1064 nm). These parameters are now included in the revised manuscript.

The sample size is too small (only 5 worms) and the difference is not so significant for figure4 c and d, on which the whole story of increased lipid mobilization is based. The authors should also elaborate how the quantification here was done, as it seems that SRS intensity was measured in the whole worm that may contain organs such as ovaries that are rich in lipid in a certain life stage.

Response: We appreciate the reviewer's suggestion, and have increased the sample size and included an independent experimental replicate in the Supplementary Figure 6. As suggested by the reviewer, we have also included detailed quantification methods in the revised manuscript.

And what's interesting is how the single amino acid mutation (rax3, rax5, rax10) will cause a dramatic change in lipid distribution, which is completely missing in the article.

Response: The single amino acid mutations are in the functionally important and conserved domains of the proteins. As shown in Figure 3a, *rax5* mutation changes glycine to arginine in 373 AA position of SMA-2; *rax10* mutation changes cysteine to tyrosine in 198 AA position of SMA-4; *rax3* mutation changes cysteine to tyrosine in 253 AA position of SMA-4. The single amino acid mutation would affect the corresponding protein function and lead to defect in the BMP signaling pathway. As characterized in the manuscript, the BMP signaling pathway regulates fat storage by controlling mitochondrial dynamics and beta-oxidation.

Reviewers' comments:

Reviewer #1 (Remarks to the Author):

This is a very strong paper on BMP signaling and Lipid metabolism. Already the initial submission was very strong. One question was whether there could be developmental effects of the BMP mutants. The authors have addressed this point full satisfactory. In my opinion, this paper should be published in its current form.

Reviewer #2 (Remarks to the Author):

The revised manuscript addressed most of the referee questions well. However, a key experiment is missing. The authors showed that expressing *sma-3* in the pharynx or in the hypodermis can rescue the mitochondrial morphology defects in *sma-3* mutant (Figure 6), suggesting that *sma-3* and therefore BMP signaling functions cell non-autonomously to regulate mitochondrial dynamics in the intestinal cells. The authors should perform similar rescue experiment using the intestinal promoter-driven *sma-3* (expressing it in the same tissue where the mitochondrial defect is observed). This should be a simple experiment to do because it appears that the authors have the plasmid in hand (Figure 3).

A minor point. In Figure 2, the authors used DWA and DWB to mark the two different conserved motifs in SMA-2 and SMA-4. This terminology is no longer used, instead, the field commonly uses MH1 and MH2 to refer to the two conserved motifs in SMAD proteins. Also, the two motifs in *sma-3* should be highlighted in the figure as well.

Reviewer #3 (Remarks to the Author):

The authors did a good job addressing my previous concerns. The revision should be publishable now.

Point-by-point response to the comments of the reviewers

Reviewer #1

This is a very strong paper on BMP signaling and Lipid metabolism. Already the initial submission was very strong. One question was whether there could be developmental effects of the BMP mutants. The authors have addressed this point full satisfactory. In my opinion, this paper should be published in its current form.

Response: We appreciate the reviewer's positive comment and his/her recommendation for publication.

Reviewer #2

*The revised manuscript addressed most of the referee questions well. However, a key experiment is missing. The authors showed that expressing *sma-3* in the pharynx or in the hypodermis can rescue the mitochondrial morphology defects in *sma-3* mutant (Figure 6), suggesting that *sma-3* and therefore BMP signaling functions cell non-autonomously to regulate mitochondrial dynamics in the intestinal cells. The authors should perform similar rescue experiment using the intestinal promoter-driven *sma-3* (expressing it in the same tissue where the mitochondrial defect is observed). This should be a simple experiment to do because it appears that the authors have the plasmid in hand (Figure 3).*

Response: We appreciate the reviewer's suggestion. We have conducted this experiment and found that the restoration of *sma-3* expression in the intestine of *sma-3* mutants partially suppresses the increased mitochondrial tubular morphology (new Fig. 6b). This result is consistent with the partial effect that the intestinal rescue confers on the fat storage level.

*A minor point. In Figure 2, the authors used DWA and DWB to mark the two different conserved motifs in SMA-2 and SMA-4. This terminology is no longer used, instead, the field commonly uses MH1 and MH2 to refer to the two conserved motifs in SMAD proteins. Also, the two motifs in *sma-3* should be highlighted in the figure as well.*

Response: As suggested by the reviewer, we have changed the motif labels to MH1 and MH2. However, for the *sma-3* allele, the mutation causes an alternative splicing change at the DNA level, with no amino acid alternation. Therefore, showing protein motifs for this allele will be inappropriate.

Reviewer #3

The authors did a good job addressing my previous concerns. The revision should be publishable now.

Response: We thank the reviewer for the positive comment and his/her recommendation for publication.

REVIEWERS' COMMENTS:

Reviewer #2 (Remarks to the Author):

The revised manuscript has successfully addressed my question.

Point-by-Point Responses

Reviewer #2 (Remarks to the Author): The revised manuscript has successfully addressed my question.

Response: We truly appreciate the positive comment of reviewer #2.